# Hollow Protein Fibers Templated Synthesis of Pt/Pd Nanostructures with Peroxidase-like Activity

**DOI:** 10.3390/v17121627

**Published:** 2025-12-16

**Authors:** Beizhe Huang, Mengting Fan, Yuhan Li, Ting Zhang, Jianting Zhang

**Affiliations:** 1Institute of Biomaterials and Tissue Engineering, Huaqiao University, Xiamen 361021, China; 2Fujian Provincial Key Laboratory of Biochemical Technology, Huaqiao University, Xiamen 361021, China

**Keywords:** tobacco mosaic virus, protein template, mineralization, metallic nanowires, peroxidase-like activity

## Abstract

Supramolecular proteins have emerged as promising templates for guiding metal ion mineralization into well-defined nanomaterials because of their structural versatility and chemical diversity. However, the precise control of metal ion nucleation on the different reactive sites of protein templates remains challenging. In this study, a genetically engineered hollow tobacco mosaic virus protein fiber (TMVF) with excellent structural stability was employed to achieve selective mineralization of noble metal nanostructures either on its external surface or within its internal channel. Moreover, the Pt/Pd bimetallic nanowire (NW) was also successfully prepared by co-depositing Pt and Pd on the TMVF. The bimetallic NWs demonstrated a peroxidase-like activity, which enabled their application for cholesterol detection by cooperating with cholesterol oxidase.

## 1. Introduction

Template-guided mineralization has emerged as an attractive strategy for preparing functional materials owing to its high controllability and designability, especially when utilizing biomacromolecules as versatile biotemplates [1,2,3]. Under the guidance of engineered biotemplates, inorganic ions can selectively nucleate and grow into well-defined architectures, thereby enabling precise control over a material’s spatial position [4], orientation [5], and composition, and facilitating the creation of functional materials with novel physicochemical properties [6,7,8].

Protein is one such biomolecule that possesses a well-defined three-dimensional structure and abundant reactive groups, offering unique advantages for controlling the size and morphology of mineralized materials [9,10,11,12,13]. Many reported studies have successfully demonstrated different protein cages, such as ferritin [14], cowpea chlorotic mottle virus [15], and simian virus 40 [16], which have been utilized to encapsulate or associate with functional nanoparticles (NPs) either within their internal cavities or on their external surfaces via in situ mineralization or co-assembly strategies. Similarly, protein fibers such as helical amyloid fibrils and phage M13 have also been employed to construct types of metal nanochains or nanowires (NWs) [17,18,19,20]. However, unlike conventional small-molecule ligands, supramolecular proteins usually possess intricate yet fragile structures, making the selective mineralization of inorganic ions on protein templates still highly challenging [21,22,23].

In this work, we demonstrate a series of one-dimensional (1D) Pd, Pt, and bimetallic nanostructures for mineralization templated by an engineered hollow tobacco mosaic virus fiber (TMVF). The disulfide bonds introduced between the TMV subunits endow the TMVF with excellent structural stability, allowing it to withstand high ambient temperatures without structural deformation (>70 °C) [24]. Metal precursors, such as Pt^2+^ or Pd^2+^, can be directly deposited on the external surface of TMVF scaffolds by simple heating, resulting in Pd-, Pt-, or bimetal-coated TMVFs with a controllable metal layer density. The confined channel of the TMVF was further utilized for the site-specific mineralization of Pd nanowires (NWs) via cyclic ascorbic acid reduction [25]. We found that Pd precursors efficiently grew into micron-sized NWs with a diameter of approximately 5 nm in the TMVF channels. Moreover, these robust TMVFs also enabled the synthesis of Pd NWs coated with Pt nanoparticles through further mineralization. The resultant noble bimetallic NWs exhibited a peroxidase-like activity and showed a possibility for cholesterol detection by cooperating with cholesterol oxidase.

## 2. Materials and Methods

### 2.1. The Preparation of Ultra-Long TMVF Template

The *E. coli* variant containing the T103C-TMV-E50C-A74C expression plasmid was kindly provided by Wang’s group at the Suzhou Institute of Nano-Tech and Nano-Bionics, Chinese Academy of Sciences. The expression and purification of the TMV coat protein were performed according to previously reported methods. Purified TMV coat proteins (5 mg/mL) were incubated in phosphate buffer (PB, 50 mM, pH 7.0) overnight, and then 50 μM CuCl_2_ was added to further promote TMVF assembly. After 24 h of incubation, the sample was centrifuged at 12,000 rpm for 10 min. The precipitate was redispersed in PB (50 mM, pH 8.0) for subsequent mineralization.

### 2.2. The Mineralization of TMVF-Templated Metallic Nanostructures

For the metal deposition on the external surface, Na_2_PdCl_4_ or Na_2_PtCl_4_ (0.1–1 mM) was mixed with 0.02 mg/mL TMVF solution and then incubated at 70 °C for 1 h. All samples were centrifuged at 12,000 rpm for 10 min, and the brown precipitate was washed with ddH_2_O three times. For the Pd NW mineralization in the inner channel of TMVF, the prepared TMVF was dialyzed against ddH_2_O and diluted to 0.1 mg/mL, then mixed with 1 mM Na_2_PdCl_4_ and subjected to ultrasonication for 30 s. After incubation for 10 min, 0.1 mg/mL PVP30K was added and sonicated for another 30 s, followed by addition of 1 mM freshly prepared ascorbic acid and further sonication for 30 s before incubation at 20 °C or 35 °C for 30 min. To obtain longer Pd NWs, after incubation for 30 min at 35 °C, ascorbic acid was added again and incubated for complete growth. For the synthesis of Pt NP–coated Pd NWs, the prepared Pd NWs underwent the same Pt deposition procedure.

### 2.3. The Measurement of Peroxidase-like Activity of TMVF-Templated Pt/Pd NWs

As a typical reaction, 20 μL of Pt/Pd NWs (1 mg/mL), 100 μL of 5 mM 3,3′,5,5′-Tetramethylbenzidine (TMB), and 100 μL of H_2_O_2_ (1 M) were added to 2.78 mL of acetic acid buffer (pH 4.0). After 2 min of reaction, the sample was analyzed using a UV–Vis spectrophotometer. For kinetic analysis using H_2_O_2_ as the substrate, 20 μL of Pt/Pd NWs (1 mg/mL) and various concentrations of H_2_O_2_ (1, 2, 4, 6, 8, and 10 mM) were added. Conversely, for kinetic analysis using TMB as the substrate, the peroxidase-like activity was evaluated by adding 30 μM H_2_O_2_ and different concentrations of TMB (0.05, 0.1, 0.2, 0.3, 0.4, and 0.5 mM) for a reaction time of 2 min.

### 2.4. Pt/Pd NWs Assisted Cholesterol Detection

In the detection of cholesterol, 50 μL of ChOx (2 mg/mL) and 100 μL of cholesterol solutions with different concentrations (50, 100, 200, 400, 600, and 800 μM) were mixed and incubated at 37 °C for 30 min. Then, 50 μL of Pt/Pd NWs (1 mg/mL), 100 μL of TMB (5 mM), and 2.7 mL of acetic acid buffer were added to the mixture. After another 30 min of incubation, the sample was analyzed using a UV–Vis spectrophotometer.

## 3. Results

TMVFs were obtained using a TMV variant with cysteine mutations at the 103rd, 53rd, and 74th residues, as recently reported by Wang’s group [24]. The introduced cysteines form disulfide (S–S) covalent bonds, which induce the TMV subunits to assemble into ultralong hollow protein fibers with excellent structural stability. Transmission electron microscopy (TEM) clearly revealed the highly efficient assembly of TMVFs after just one day of incubation in phosphate buffer (Figure 1A,B). These assembled TMVFs exhibited desirable thermostability, maintaining structural integrity without any breakage after incubation in water at 70 °C (Appendix A). More importantly, the abundant ctableharged groups on their external surfaces facilitate metal ion deposition [26,27]. Specifically, Pd^2+^ or Pt^2+^ ions spontaneously nucleated and grew into small NPs on the surface of TMVFs after 1 h of co-incubation at 70 °C. The density of the NPs could be controlled by concentration or by tuning the reaction time (Figure 1C,D and Appendix A). Notably, Pd^2+^ or Pt^2+^ ions have a different nucleation and growth rate on the TMVF. 0.2 mM Pt^2+^ ions would nucleate and grow into 2~3 nm NPs (Appendix A). The density of NP would increase with the increasing Pt^2+^ in the reaction system (Appendix A). Compared with the Pt^2+^ ion, Pd^2+^ undergoes quicker nucleation and growth. We could find that dense Pd NPs form on the TMVF even in a low reaction concentration of 0.2 mM Pd^2+^ (Appendix A). When Pd^2+^ increases in the reaction system, the TMVF would be coated by a thick Pd shell with a thickness of several nanometers (Appendix A). In addition, bimetallic Pt/Pd NWs were successfully obtained through simple co-growth (Figure 1F and Appendix A). Energy-dispersive spectroscopy (EDS) mapping revealed that the resulting NWs consisted of Pt and Pd elements with uniform distribution (Figure 1G–I).

In addition to the external surface of TMVFs, the inner channel also served as another desirable reactive site for the confined mineralization of novel metals [28]. Although earlier studies have reported the mineralization of metal nanochains or NWs in the channels of TMV nanorods, their yields were limited by the structural instability of native TMV templates [29,30,31]. In contrast, our robust TMVFs significantly improved the synthetic efficiency of Pd NWs within their inner channels (Figure 2A,B). In detail, Pd^2+^ ions were mixed with TMVFs and polyvinylpyrrolidone (PVP30K) and then reduced by ascorbic acid under magnetic stirring. PVP30K has been shown to enhance Pd nucleation in the inner channel of TMV fiber. TEM images revealed that numerous short Pd nanorods rapidly formed in the cavities of TMVFs (Figure 2C,F). The growth of the Pd nanorods was strongly influenced by the reaction temperature. When the temperature increased from 25 °C to 37 °C, the length of the Pd NWs remarkably increased from tens of nanometers to several hundred nanometers (Figure 2D,G). It is worth noting that Pd growth did not cause any breakage of the TMVF template. Therefore, a cyclic growth strategy was adopted to achieve high-quality NWs. After two cycles of growth at 37 °C, most of the Pd NWs reached several micrometers in length (Figure 2E,H). The confined channels ensured that the Pd NWs consistently maintained a diameter of approximately 5 nm, even as their length increased from several nanometers to several micrometers. The yield and quality of the resulting Pd NWs were greatly improved compared with previously reported TMV-based mineralization systems, which can be attributed to the excellent structural stability of the TMVFs. However, no similar NW formation was observed in the TMV channels when using other metal ions such as Pt^2+^ or Ag^+^, which instead randomly formed nanoparticles in solution (Appendix A). We speculated that the reduction potential of Pd^2+^, Pt^2+^ and Ag^+^ is the key factor in determining the nucleation and growth in the channel or in the solution. Moreover, the pH value, temperature, and reducing agent also affect the procedure. In general, Pd^2+^ is more easily nucleated in the channel of TMV than other metal ions based on this synthetic method.

This multifunctional TMVF template also enabled the construction of Pd NW within the inner channel and Pt NPs on the external surface together. To this end, the prepared Pd NWs were further deposited by the Pt precursors on the external surface. As shown in Figure 3A, the low-magnification TEM image demonstrated that the Pd NWs retained their ultralong structure without any fracture after incubation at 70 °C. Dense Pt NPs were clearly observed on the TMVF surface, forming Pt-coated Pd NW structures (Figure 3B). EDS mapping confirmed that the resulting composite NWs were composed of Pt and Pd elements (Figure 3C).

Pd- or Pt-based nanomaterials have been verified to possess excellent peroxidase-like activity [32,33]. The catalytic performances of TMVF-templated Pt, Pd and their bimetallic NWs were investigated by catalyzing the oxidation of TMB in the presence of H_2_O_2_. Pt/Pd bimetallic NW exhibited the best catalytic performance compared with Pt or Pd NWs (Appendix A). Also, the Pt@Pd NW had a lower catalytic activity than the Pt/Pd NW. We speculated that the catalytic efficiency of the inner Pd NW was restricted by the TMVF template. Therefore, only Pt/Pd NW was chosen as the catalyst for the cholesterol test. Here, we carefully investigate the catalytic activity of the optimized TMVF-templated Pt/Pd NWs. In a typical process, 20 μg of Pt/PdNW catalyst was added to 3 mL of sodium acetate buffer (0.1 M, pH 5.5) containing 100 μL of TMB (5 mM) and 100 μL of H_2_O_2_ (1 M). As shown in Figure 4A,B, the absorbance of the Pt/Pd NWs + TMB + H_2_O_2_ system (red line) at 652 nm was greatly enhanced compared with the control groups and increased continuously with reaction time. The peroxidase-like activity of Pt/Pd NWs also shows a positive correlation with the density of NPs (Appendix A). These results confirmed the intrinsic peroxidase-like activity of the Pt/Pd NWs. To obtain the steady-state kinetic parameters, the enzymatic reaction kinetics were monitored in systems containing different concentrations of TMB or H_2_O_2_. As shown in Figure 4C,D, typical Michaelis–Menten curves and double-reciprocal plots of the initial reaction rate (insets of Figure 4C,D) were obtained when using either TMB or H_2_O_2_ as the substrate. The Michaelis constant (Km) of the Pt/Pd bimetallic NWs was calculated to be 0.11 × 10^−3^ M for TMB and 9.57 × 10^−3^ M for H_2_O_2_. Compared with previously reported data, the Km values of the Pt/Pd NWs toward TMB and H_2_O_2_ were significantly lower than those of HRP, indicating that the Pt/Pd bimetallic NWs exhibit stronger substrate affinity toward both TMB and H_2_O_2_ (Appendix A).

The peroxidase-like activity endowed the TMVF-templated Pt/Pd NWs with the capability for cholesterol detection in cooperation with cholesterol oxidase (ChOx). Cholesterol was catalyzed by ChOx in the presence of O_2_ to generate 4-cholesten-3-one and H_2_O_2_, which subsequently oxidized TMB to produce blue oxTMB under the catalysis of the Pt/Pd NWs. As shown in Figure 5A, the absorbance value clearly increased with cholesterol concentration ranging from 0 to 800 μM. When the cholesterol concentration was between 50 and 800 μM, the absorbance intensity increased linearly with the cholesterol concentration (Figure 5B). The selectivity of the Pt/Pd NWs-based sensing system was also evaluated. The absorbance responses toward potential interferents, including glucose, serine, dopamine, cysteine, ascorbic acid, lysine, histidine, and arginine (each at 10 mM), were monitored and compared with that of cholesterol. The addition of cholesterol resulted in a fivefold increase in absorbance at 650 nm. In contrast, no obvious change in absorbance was observed when other interferents were added. In conclusion, the Pt/Pd NWs-based sensing system exhibited excellent sensitivity and satisfactory selectivity toward both H_2_O_2_ and could be applied to cholesterol detection under the assistance of cholesterol oxidase.

## 4. Conclusions

We demonstrated a series of Pd, Pt and their bimetallic NW mineralizations using ultralong and robust TMVF as a template. The hollow structural feature and excellent structural stability of TMVFs endow them with the ability to direct the selective nucleation of Pd ions either within the inner channel via ascorbic acid reduction or on the external surface through thermal incubation. Compared with previously reported TMV-based metallic mineralization processes, which often suffer from low yield and poor quality, our results demonstrate the highly efficient synthesis of microscale metallic NWs. Furthermore, the optimized Pt/Pd NWs exhibited peroxidase-like activity, enabling their use for cholesterol detection under the assistance of cholesterol oxidase.

## Figures and Tables

**Figure 1 viruses-17-01627-f001:**
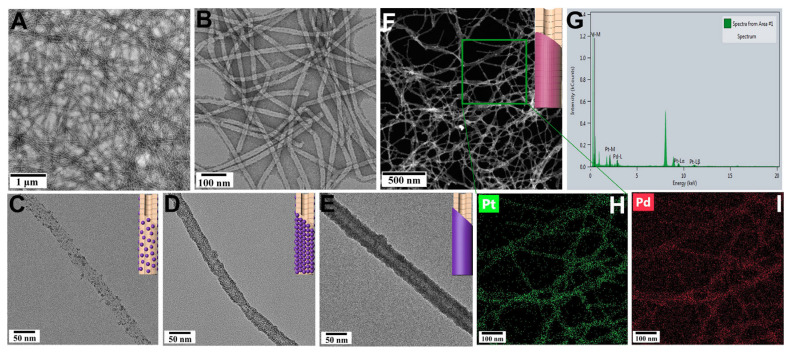
TEM characterization of engineered TMVFs and mineralization on their external surfaces. (**A**,**B**) TEM images of ultralong TMVFs. (**C**–**E**) TEM images showing Pt deposition on the external surface of TMVFs with increasing nanoparticle density. (**F**–**I**) HAADF-STEM image (**F**); corresponding EDS spectrum (**G**); and elemental mapping images (**H**,**I**) of bimetallic Pt/Pd NWs.

**Figure 2 viruses-17-01627-f002:**
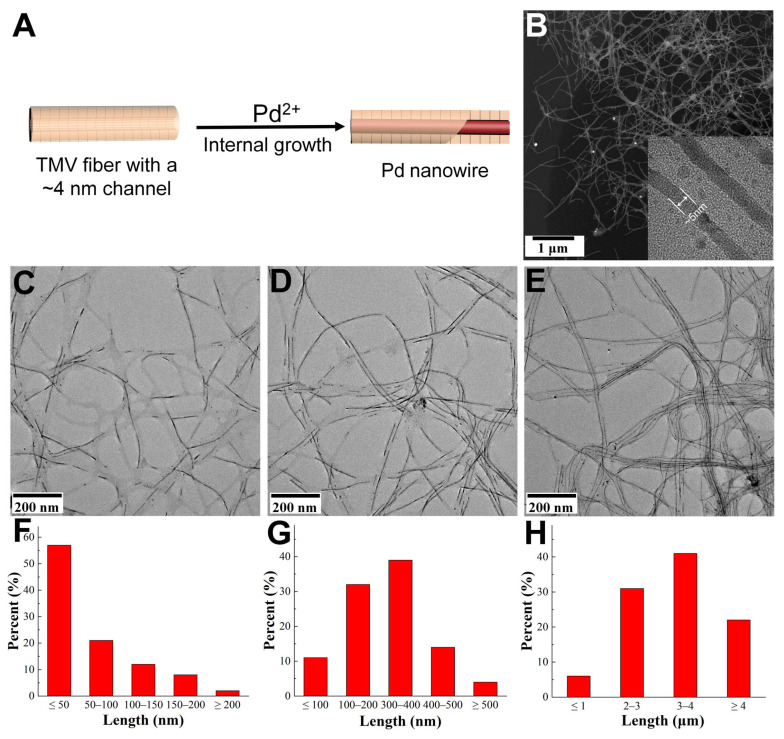
Preparation of Pd NWs in the inner channels of TMVFs. (**A**) Schematic illustration of the Pd NW mineralization process. (**B**) Low-magnification TEM image of the resulting slender Pd NWs with an average diameter of ~5 nm. (**C**–**E**) TEM images of TMVF-templated Pd NWs synthesized via ascorbic acid reduction at 25 °C (**C**), 37 °C (**D**), and 37 °C with two growth cycles (**E**). (**F**–**H**) Corresponding statistical histograms of the Pd NW lengths.

**Figure 3 viruses-17-01627-f003:**
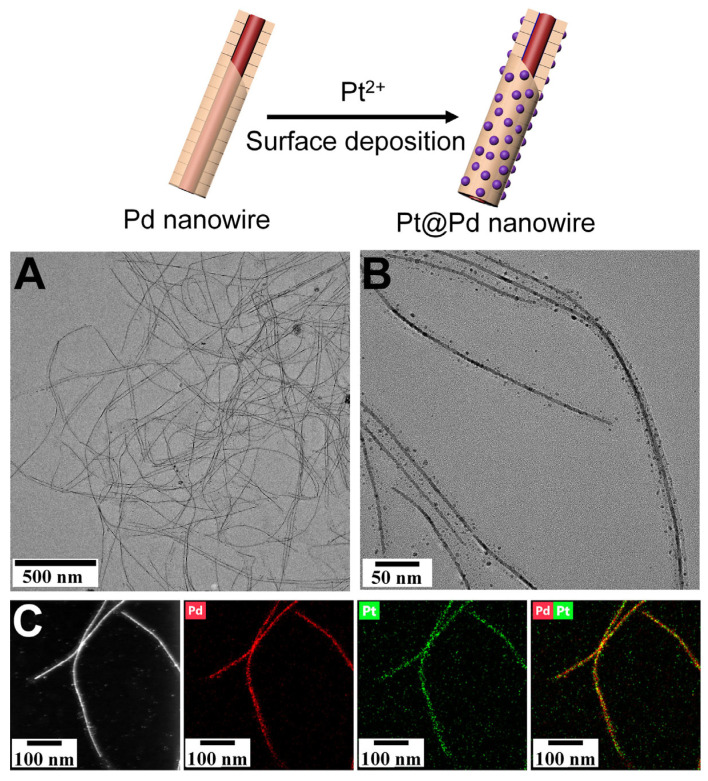
TEM characterization of the TMVF-templated composite Pt-coated Pd NWs. (**A**,**B**) TEM images of microscale Pd NWs coated with Pt NPs. (**C**) HAADF-STEM images and corresponding EDS elemental mappings of the Pt-coated Pd NWs.

**Figure 4 viruses-17-01627-f004:**
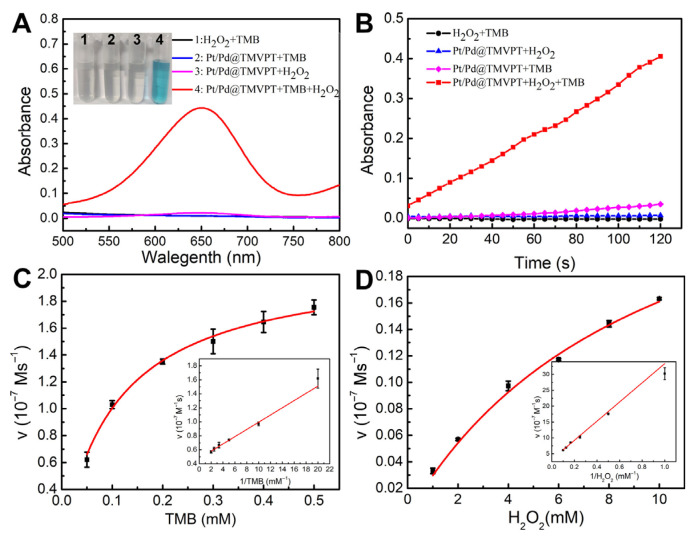
Steady-state kinetic assays of the TMVF-templated Pt/Pd NWs. (**A**) UV–vis absorbance spectra of TMB in different reaction systems. Inset: typical photographs of the corresponding reaction systems. (**B**) Time-dependent absorbance changes at 650 nm for various reaction systems. (**C**,**D**) Steady-state kinetic assays of the TMVF-templated Pt/Pd NWs: (**C**) 33.3 mM H_2_O_2_ with varying concentrations of TMB; (**D**) 0.17 mM TMB with different concentrations of H_2_O_2_. Insets: Double-reciprocal plots corresponding to the Pt/Pd NWs catalyst, respectively.

**Figure 5 viruses-17-01627-f005:**
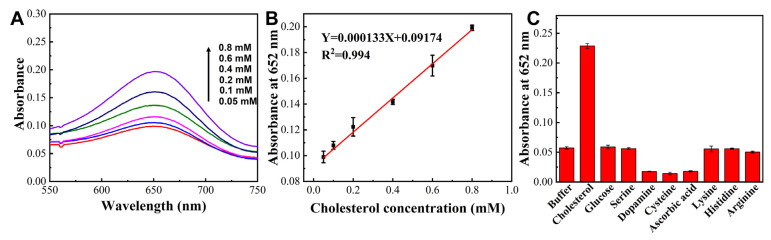
(**A**) Absorbance response profiles of the Pt/Pd NWs-based fluorescent sensor for different concentrations of cholesterol. (**B**) Linear calibration curve for cholesterol concentrations. (**C**) Specificity of the sensing system toward cholesterol (1 mM) and various interfering substances (10 mM).

## Data Availability

The original contributions presented in this study are included in the article. Further inquiries can be directed to the corresponding author.

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
