# Peer review of "Hollow Protein Fibers Templated Synthesis of Pt/Pd Nanostructures with Peroxidase-like Activity"

_viruses, 2025, doi:10.3390/v17121627_

Round 1
Reviewer 1 Report
Comments and Suggestions for Authors
Hollow Protein Fibers Templated Synthesis of Pd/Pt Nanostructures with High Peroxidase-Like Activity
By Beizhe Huang, Mengting Fan, Yuhan Li, Ting Zhang, Jianting Zhang
In this article, the authors are using an engineering Tobacco mosaic virus to fabricate bimetallic Pd/Pt nanowires. The newly synthetized nanohybrid was used as catalysis for peroxidase-like activity. The experimental results are conclusive, but explanations regarding the synthesis, experimental conditions, controls, and a more detailed discussion are lacking. I don't think the article is ready to be published in Viruses yet.
Majors remarks
- pH values must be specified for the mineralization experiments. This is an important parameter which can explain external or internal mineralization of TMVF.
- What is the role of PVP30K? Why no Pd nanostructure was observed outside TMVF when Pd NWs in the inner channel was synthetized. It seems more easy to reduce Pd2+ on the external surface (without any reductant agent).
- Why Pt2+ and Ag+ can not enter the channel? Or be reduced inside? Is it a question of charge, potential or something else?
- Lane 104, the authors said “the density of NPs can be controlled…” but, they never specify the density and the Np size of their synthetized nanohybrid used for peroxidase activity. These two parameters are important for catalysis experiments.
- I don’t understand the role of Pd nanowire inside de TMVF in the peroxidase reaction. It is needed to achieve this catalytic reaction? Do the authors only test with Pt nanostructures outside TMVF? If the Pd nanowire is necessary, this should be discussed.
- For the detection of cholesterol, in which medium is it performed? Phosphate buffer? In the presence of detergent?
- Lane 178, TMB and H2O2 have good affinity for Pd/Pt NWs, but the Vmax values are much lower than those from table S1. It is not discussed. Maybe, the authors should compare the kcat values.
Minors remarks
- In chemical formulas, numbers must be subscripted, and the charge of cations must be superscripted (example lines 44 and 64).
- T103C-TMV50/74C should be designed as T103C-TMV-E50C-A74C to explain the mutations.
- Figure3 Pd@Pt NWs should be replaced by Pd/Pt NWs for homogeneity
- TMB should be explain at least once as 3,3ʹ,5,5ʹ-Tetramethylbenzidine.
- Line 175 et 176, the KM values are reversed.
Reviewer 2 Report
Comments and Suggestions for Authors
The authors claim to have created a series of selective one-dimensional (1D) materials. There is no evidence that these materials are selective only for Pd and Pt. The authors need to either prove selectivity using other divalent cations or remove any discussion of selectivity from the text.
The authors write about the formation of Pt or Pd nanowires on the outer surface or inner surface of a protein channel. What evidence do the authors have that nanowires are formed? What evidence do the authors have that Pt deposits form on the outer surface, not the inner surface? Electron microscopy with elemental analysis is not proof. The criterion for a nanowire is not discontinuity and conductivity. Can the authors provide evidence?
Why do the authors believe their metal samples exhibit peroxidase-like activity? What evidence is there? If you simply add divalent platinum or palladium to the system used by the authors, the same thing will be observed. And this has long been known! In this case, divalent metals (+2) will convert to tetravalent metals (+4).
The authors' arguments about a selective cholesterol probe don't hold up to scrutiny. It's not even that a single fat-soluble compound is being compared with many water-soluble ones, although this in itself is strange (again, this applies to the word "selective"). The test system, in the presence of almost 1 mM cholesterol (an enormous concentration), shows a signal four times greater than the control. Only signal than order of magnitude greater than the original can be used as proof of a true measurement.
And the objectives of the study are not clearly defined. This needs to be corrected!
Overall, it's not clear to me what new has been achieved. It's obvious that proteins contain many cysteine ​​residues, meaning reducing equivalents exist and are possibly isolated from the bulk by the protein's geometry. The authors then add Na2PdCl4 or Na2PtCl4 salts to the reducing agent and observe the formation of nanostructures. What's new about this? Is it better than anyone else's? Or better than the authors' previous results? Furthermore, I'm concerned that the manuscript doesn't provide definitive evidence that we're not observing simple chelation and mineralization with substitution, for example, of sodium and potassium in the protein structure (It's usually left over from a buffer solution).
The authors use multiple references to support a single sentence in text. If these references support the same phenomenon, include only one reference. If a sentence contains multiple facts that support different references, place the references after each fact, not at the end of the sentence.
Comments on the Quality of English LanguageI had difficulty understanding the meaning of a significant number of sentences in the manuscript.
Round 2
Reviewer 1 Report
Comments and Suggestions for Authors
The authors have revised the manuscript taking into account the comments from the referees. However, the original concerns raised by this referee is still not addressed.
I now understand that the catalysis experiments were performed on the Pt/Pd NWs, co-deposited on TMVF. However there is still no control group using only Pt or Pd. The impact of co-deposition is not clear on the performance of catalysis.
What is the purpose of Pt@Pd nanowire? The authors did not perform the catalysis experiments with this object.
We have no information regarding density and Np size of their nanostructures. However, I believe that these parameters are important for catalysis and can be obtained from TEM images.
Minor remarks
In the abstract, line 16, “sensitive fluorescence sensors”, should be replaced by “sensitive absorbance sensors”.
Line 100, Specifically, with a p
Line 102, A word is missing in the sentence “…could be controlled by…”
Line 140, the reference is missing
Line 167, there is no figure S4 in the supplementary information.
Figure 5. It is not fluorescence response, but absorption response.

Reviewer 2 Report
Comments and Suggestions for Authors
The authors have improved the manuscript, but it still contains statements and assertions unsupported by facts. 1. The authors have not provided convincing evidence that their metal samples exhibit peroxidase-like activity. What evidence is there? My previous passage about divalent platinum or palladium was an example; the same thing will occur when exposed to metallic (zero-valent) platinum or palladium. In this case, zero-valent metals (0) are converted to divalent metals (+2). Basically, this is analogous to the well-known Fenton reaction. To prove any catalytic activity, the authors need to prove the immutability of their nanoobjects, both in mass and charge. There is no evidence that the nanoobjects are immutable in both mass and charge! 2. The selectivity for cholesterol is very low, making it impossible to talk about creating a probe or other selective agent. The selection of compounds to illustrate the idea does not stand up to scrutiny. Moreover, some compounds reduce the signal level, meaning that the resulting signal will differ significantly when combined. If we're talking about cholesterol being catalyzed by cholesterol oxidase in the presence of Oâ‚‚ to form 4-cholesten-3-one and Hâ‚‚Oâ‚‚, which are then oxidized by TMB, forming TMB blue ox under the action of Pd/Pt nanofibers, then there's no selectivity here... Basically, the authors need to prove their point or lower the level of their claims.
Comments on the Quality of English LanguageI had difficulty understanding the meaning of a significant number of sentences in the manuscript.
Round 3
Reviewer 1 Report
Comments and Suggestions for Authors
The authors have revised the manuscript taking into account the comments from the referees. I believe this work can be published now in Viruses.
I have however a minor remark: I think a sentence should be added to the text explaining why the beautiful objects obtained (TMVF-templated composite Pt-coated Pd NWs) are not used for catalysis, as explained by the authors in their response in my comment 2.
Author Response
Comment: I have however a minor remark: I think a sentence should be added to the text explaining why the beautiful objects obtained (TMVF-templated composite Pt-coated Pd NWs) are not used for catalysis, as explained by the authors in their response in my comment 2.
Response: Thanks the reviewer's suggetion. A sentence has been added to the text to explain why we used the Pd/Pt NW instead of Pd@Pt NW for the cholesterol detection in the revised manuscript.
Reviewer 2 Report
Comments and Suggestions for Authors
Unfortunately, after two rounds of review, the authors failed to address my concerns. They didn't conduct any additional experiments. Unfortunately, the analogy method the authors use doesn't constitute proof. Just because other authors have proven their material has catalytic properties doesn't mean a similar material also has the same properties.
Comments on the Quality of English LanguageI had difficulty understanding the meaning of a significant number of sentences in the manuscript.
Author Response
Comment: Unfortunately, after two rounds of review, the authors failed to address my concerns. They didn't conduct any additional experiments. Unfortunately, the analogy method the authors use doesn't constitute proof. Just because other authors have proven their material has catalytic properties doesn't mean a similar material also has the same properties.
Response: Thanks your comment. We are sorry that we can't dispel his/her doubt about the catalytic properties of NWs. The XPS result (in the last response letter) has revealed that the prepared TMVF-templated Pt NWs was zero-valent Pt nanowire, which still exhibited the catalytic effect for TMB. Therefore, it's reliable that Pt NW has the peroxidase-like activity.